



# El Niño, La Niña, and the global sea level budget

Christopher G. Piecuch[1] and Katherine J. Quinn[1]

[1]Atmospheric and Environmental Research, Inc., Lexington, MA 02421, USA.

*Correspondence to:* Christopher G. Piecuch (cpiecuch@aer.com)

**Abstract.** Previous studies show that nonseasonal variations in global-mean sea level (GMSL) are significantly correlated with El Niño-Southern Oscillation (ENSO). However, it has remained unclear to what extent these ENSO-related GMSL fluctuations correspond to steric (i.e., density) or barystatic (mass) effects. Here we diagnose the GMSL budget for ENSO events observationally us-

ing data from profiling floats, satellite gravimetry, and radar altimetry during 2005–2015. Steric and barystatic effects make comparable contributions to the GMSL budget during ENSO, in contrast to previous interpretations based largely on hydrological models, which emphasize the barystatic component. The steric contributions reflect changes in global ocean heat content, centered on the Pacific. Distributions of ocean heat storage in the Pacific arise from a mix of diabatic and adiabatic effects.

Results have implications for understanding the surface warming slowdown and demonstrate the usefulness of the Global Ocean Observing System for constraining Earth's hydrological cycle and radiation imbalance.

## 1 Introduction

Sea level is an informative index of climate and serious concern for coastal communities. Hence, un-

derstanding the modern altimetry record is important from scientific and societal vantage points. The most apparent signals in the altimetric global-mean sea level (GMSL) data are the annual cycle and linear trend (e.g., Figure 4 in Masters et al., 2012). In principle, these changes in the global ocean's water volume relate to the ocean's mass and its density, referred to as 'barystatic' and 'steric' sea level changes, respectively (e.g., Gregory et al., 2013; Leuliette, 2015). Past studies have successfully

used *in situ* hydrography and satellite gravity data to assess ocean mass and density changes and to evaluate barystatic and steric effects on the annual cycle and the linear trend in GMSL (e.g., Lom-



bard et al., 2007; Willis et al., 2008; Cazenave et al., 2009; Leuliette and Miller, 2009; Leuliette and Willis, 2011; Leuliette, 2014, 2015).

Although the annual cycle and linear trend are the most prominent signals in the record, altimeter data also evidence more subtle GMSL variations superimposed on those signals. In particular, it has long been reported that nonseasonal GMSL anomalies are significantly correlated with El Niño-Southern Oscillation (ENSO), such that the GMSL is anomalously positive during warm El Niño phases and anomalously negative during cool La Niña phases (Nerem et al., 1999; Chambers et al., 2002; Ngo-Duc et al., 2005; Landerer et al., 2008; Merrifield et al., 2009; Llovel et al., 2010; Nerem et al., 2010; Llovel et al., 2011; Boening et al., 2012; Cazenave et al., 2012; Meyssignac and Cazenave, 2012; Stammer et al., 2013; Fasullo et al., 2013; Haddad et al., 2013; Meyssignac et al., 2013; Calafat et al., 2014; Cazenave et al., 2014; Dieng et al., 2014; Pugh and Woodworth, 2014; Dieng et al., 2015). Recent papers argue that ENSO-related GMSL changes are essentially of barystatic origin, related to changes in the hydrological cycle, and patterns of precipitation and evaporation (Llovel et al., 2011; Boening et al., 2012; Cazenave et al., 2012, 2014; Fasullo et al., 2013). However, these papers are based on either observations during an isolated event or models, and the extent to which barystatic or steric effects are responsible for ENSO-related GMSL fluctuations more generally has not been firmly established based on observations. In fact, conflicting accounts of the GMSL budget during ENSO events are given in the literature. For example, based on altimetry, sea-surface temperature data, and ocean model output, Nerem et al. (1999) reason that the anomalous GMSL rise during the 1997–1998 El Niño was due to thermal expansion of the upper ocean. In contrast, using altimetry and global hydrological models, Ngo-Duc et al. (2005), Llovel et al. (2011), and Cazenave et al. (2012) argue that this anomalous rise in GMSL was owing to an increase in global ocean mass. On the one hand, based on satellite data and *in situ* observations, Boening et al. (2012) and Fasullo et al. (2013) conclude that the anomalous fall in GMSL during the 2010–2011 La Niña was related to a decrease in global ocean mass. On the other hand, and based on very similar datasets, Dieng et al. (2014) conclude differently, finding that this anomalous GMSL fall was owing in approximately equal parts to barystatic and steric contributions.

The literature thus paints a confusing portrait. Clarifying the nature of ENSO-related GMSL variations is important for understanding the ocean's role in Earth's hydrological cycle and energy imbalance (e.g., Fasullo et al., 2013; Leuliette, 2015). Here we exploit the growing record length of the Global Ocean Observing System, analyzing satellite gravity, radar altimetry, and *in situ* hydrographic observations using linear estimation (regression) to elucidate observationally the nature of the altimetric GMSL budget for ENSO events.



## 2   Datasets

### 2.1   Satellite altimetry

We study GMSL records from four groups: AVISO (Ablain et al., 2009), Colorado (Nerem et al., 2010), NOAA (Leuliette and Scharroo, 2010), and CSIRO (Church and White, 2011). Time series derive from the reference altimetry missions (TOPEX/Poseidon, *Jason*-1, -2). The standard corrections (postglacial rebound, wet troposphere, inverted barometer) are made and a 60-day filter is used to remove a spurious 59-day signal (Masters et al., 2012). Time series are interpolated onto regular monthly intervals over 1993–2015 and we use the ensemble average across the interpolated records. A standard error (Table 1) is estimated based on variances in differences between time series (cf. Ponte and Dorandeu, 2003).

### 2.2   Profiling floats

Monthly Argo *in situ* temperature and salinity grids produced by Scripps Institution of Oceanography (SIO) and International Pacific Research Center (IPRC) are also employed. The grids are generated using objective analysis applied to quality controlled float profiles (Roemmich and Gilson, 2009). Fields span from 65°S to 65°N latitudinally, and down to $\sim$ 2000 m, but do not cover marginal shelf seas. We use the data for the period 2005–2015, since float coverage was not sufficient before then (Leuliette, 2015, and references therein). We use these gridded fields to evaluate steric sea level following Gill and Niiler (1973). And as with altimetry data, we use the average of the SIO and IPRC time series, deriving a standard error using the difference between these products (Ponte and Dorandeu, 2003).

### 2.3   Gravimetric retrievals

Monthly estimates of the barystatic sea level term based on retrievals from the Gravity Recovery and Climate Experiment (GRACE) (e.g., Tapley et al., 2004) are also considered. Values are from Release-05 data processed by the three main science data system centers at CSR (Bettadpur, 2012), JPL (Watkins and Yuan, 2012), and GFZ (Dahle, 2013). These data are then postprocessed by Don P. Chambers at University of South Florida following the methods detailed in Chambers and Bonin (2012) and Johnson and Chambers (2013). We consider the ensemble mean across the estimates, deriving an estimate of the standard error according to variances in the differences between series (Ponte and Dorandeu, 2003). To be overlapping with Argo, we consider the GRACE ocean mass data over 2005–2015.





## 3 Results and discussion


Figure 1a shows nonseasonal anomalies of GMSL (i.e., annual cycle and trend removed) alongside the Multivariate ENSO Index (MEI) (Wolter and Timlin, 1998) over 2005–2015. As in earlier papers cited above, there is a tight relation between GMSL and MEI curves, such that the GMSL is higher during El Niño periods and lower during La Niña periods. The Pearson product-moment correla-

tion coefficient (hereafter simply referred to as the correlation) between these two records (0.73) is significant at the 95% confidence level and suggests that approximately half of the nonseasonal anomalous GMSL variance over this period corresponds to ENSO. More generally, we observe that correlation between the nonseasonal GMSL and MEI anomalies is significant for all other 11-year periods during the altimeter record, as well as for the entire 23-year altimetric record itself (not

shown).

Nonseasonal GMSL anomalies from satellite altimetry data are consistent with the sum of barystatic and steric components from GRACE and Argo (Figure 1b). The correlations between GMSL from GRACE and Argo and from altimetry (0.89), and between MEI and sum of GRACE and Argo (0.67) are both significant. Correlation values between GRACE and the MEI (0.54; Figure 1c) and Argo

and the MEI (0.65; Figure 1d) are also significant. In fact, all pairs of time series displayed in Figure 1 are significantly correlated (not shown). These results suggest that GMSL fluctuations tied to ENSO and seen by satellite altimetry are independently corroborated by the other ocean observing platforms and that barystatic and steric terms both contribute to the significant relationship between GMSL and ENSO.

To consider the GMSL budget related to ENSO more formally, we use linear estimation, namely ordinary least squares (OLS). We model the observations as linear combinations of decadal trend, annual cycle, and MEI regressors, simultaneously solving for the regression coefficients for all predictors by minimizing the residual. While OLS assumes the residuals behave as white noise, in practice we find that residuals are serially correlated (not shown). Thus, we inflate the standard errors

according to the lag-1 autocorrelation and the effective degrees of freedom as detailed in Chambers et al. (2012) and Calafat and Chambers (2013). More technical details of our methods can be found in Appendix A.

Table 1 shows results of this OLS procedure applied to altimetry, GRACE, and Argo. All quoted values are 90% confidence intervals as described in Appendix B. (Since they are not our focus

here, we defer discussion of the results for the annual cycle and trend to Appendix C.) Per unit MEI change, altimetric GMSL changes by $2.76 \pm 1.87$ mm, which is close to the value of $2.97 \pm 1.47$ mm given by the sum of Argo steric and GRACE barystatic terms. Indeed, the residual value is not statistically distinguishable from zero ($-0.20 \pm 0.64$ mm), showing that the GMSL budget related to ENSO can be closed using observational data. Significant regression coefficients are also

determined for Argo steric ($1.42 \pm 0.53$ mm) and GRACE barystatic ($1.54 \pm 1.50$ mm) components. The error bars on the barystatic term are comparatively wider than on the steric term, agreeing with





the relatively stronger correlation between Argo and MEI than between GRACE and MEI seen above (Figure 1).

The OLS regression coefficients demonstrate that steric and barystatic effects generally make comparable contributions to the ENSO-related GMSL changes over the study period. Judging from Monte Carlo simulations performed using values in Table 1 (see Appendix D), it is as likely as not (33–66% likelihood) that barystatic effects are responsible for 45–58% of the sum of barystatic and steric contributions to GMSL variations linked to ENSO, and very unlikely ($< 10\%$ likelihood) that the barystatic term amounts to $> 68\%$ (Figure 2). This is at odds with the emphasis placed on the barystatic contribution by recent studies (e.g., Llovel et al., 2011; Cazenave et al., 2012, 2014), revealing that, at least over this time period, the steric component is equally as important.

Regional distributions of ENSO-related terrestrial water storage, which are ultimately coupled to the barystatic contributions to GMSL fluctuations through mass conservation, are explored in past papers (Llovel et al., 2011; Boening et al., 2012; Phillips et al., 2012; Fasullo et al., 2013; de Linage et al., 2013; Eicker et al., 2016); they are not revisited here. However, ENSO-related GMSL behavior owing to steric effects is not as well understood. The steric contributions to the GMSL fluctuations related to ENSO arise from changes in ocean heat content. Arguments based on mass conservation (Munk, 2003) suggest that any global steric contributions resulting from salinity changes would be exceedingly small. To elucidate ocean heat content changes potentially contributing to GMSL changes related to ENSO we thus apply the OLS method to Argo vertical potential temperature profiles, averaging horizontally over the global ocean as well as individual ocean basins (Figure 3).

There is significant warming of the global ocean's surface waters (0–100 m) and cooling within its main thermocline (130–320 m) during El Niño periods. Marginally significant warming also occurs at some intermediate depths (600–650 m). On the whole, the global upper ocean (0–2000 m) gains $5.5 \pm 5.2$ ZJ (ZJ $\equiv 10^{21}$ J) of heat per unit of MEI increase (equivalent to a spatially uniform global ocean temperature variation of $\mathcal{O} \sim 0.001°C$). While there are some significant thermal changes related to ENSO observed in other basins at some depths ($< 60$ m in the Indian; $> 1350$ m in the Atlantic), the vertical structure of the global ocean's ENSO-related thermal variations derives from the Pacific, where there is similar warming near the surface (0–110 m), cooling in the thermocline (130–320 m), and warming of intermediate waters (500–1150 m). Indeed, only the Pacific experiences significant net thermal changes during ENSO, hardly surprising seeing as ENSO originates from coupled air-sea interactions in the Pacific (e.g., Clarke, 2008, and references therein).

Given only the Argo data, one cannot unambiguously assess heat budgets for the various layers over the different basins. One possible interpretation is that net Pacific heat storage is owing to local surface heat exchanges with the atmosphere. This interpretation assumes no contributions from the deep ($> 2000$ m) and no fluxes between basins, and demands heat fluxes from the thermocline layer to the surface and intermediate layers (Figure 4). Our interpretation is supported by Mayer et al. (2014), who argue that ocean heat storage over the tropical Pacific (30°S–30°N) during ENSO is



balanced by surface heat exchanges. Other interpretations are possible given the data, but would
imply that surface heat fluxes over every other basin are balanced and compensated by ocean heat
transports out of or into that basin. Any more definitive diagnosis of the heat budgets would require
a more advanced approach (e.g., based on a physically consistent state estimate constrained by the
available ocean observations (e.g., Forget et al., 2015)), which is beyond our scope and deferred to
future study.

Previous studies suggest that both the global ocean and climate system lose heat during El Niño
events (e.g., Roemmich and Gilson, 2011; Loeb et al., 2012; Trenberth et al., 2014). This would ap-
pear to conflict with our finding that the ocean is warmer during El Niños. However, the discrepancy
is only apparent, since we consider ocean heat content and those past studies focus on the ocean heat
content *tendency* (i.e., its rate of change). Moreover, scrutinizing visual examination of the earlier
results (e.g., Figure 8 in Trenberth et al., 2014) suggests that there is a phase lag between ENSO and
the heat content tendency, such that warming precedes El Niño peaks and cooling follows peaks.
This would be fully consistent with our findings, and those of von Schuckmann et al. (2014), who
show a negative global ocean heat content anomaly during the 2010–2011 La Niña. It would be
helpful for future studies to examine in closer detail the coherence between ocean heat content and
ENSO.

These results have implications for understanding the recent 'surface warming slowdown', which
some partly relate to the dominant La Niña phase of the 2000s relative to the 1990s (Kosaka and Xie,
2013; Cazenave et al., 2014; England et al., 2014; Risbey et al, 2014). Nieves et al. (2015) determine
that the slowdown was caused by a decadal shift in Indo-Pacific heating; they show that the Pacific
Ocean above 100 m cooled while the Indian Ocean between 100–300 m warmed from the 1990s to
the 2000s, but that the rate of global ocean heat storage above 1500 m did not change during that
time. Our results (Figure 3) suggest that cooling of the surface Pacific between the two decades is
consistent with phasing of ENSO, but subsurface Indian warming and lack of net ocean warming or
cooling are not, hinting that processes unrelated to ENSO also contributed to the surface warming
slowdown, consonant with papers showing an important role for the Interdecadal Pacific Oscillation
(Meehl et al., 2013; Trenberth and Fasullo, 2013; Steinman et al., 2015; Fyfe et al., 2016).

We note that nonseasonal anomalous GMSL was considerably higher during the 2014–2015 El
Niño than during the 1997–1998 El Niño (Figure 5), which is noteworthy because these two El
Niño events were comparable in amplitude. This could suggest that the relationship between GMSL
and ENSO is a complicated function of time period and frequency band, in which case the results
presented here apply strictly to the study period. However, it could also suggest that other climate
modes (e.g., Pacific Decadal Oscillation (e.g., Hamlington et al., 2016)) exert an influence on GMSL
that has yet to be discussed. Addressing these interesting topics is beyond our scope and left for
future investigations.





**4  Conclusions**

It has long been recognized that nonseasonal variations in global-mean sea level (GMSL) are correlated with measures of El Niño-Southern Oscillation (ENSO), but the nature of such GMSL fluctuations tied to ENSO, whether steric or barystatic, has remained unclear. We diagnosed the GMSL budget related to ENSO based on altimetry, GRACE, and Argo. Fluctuations in ENSO, GMSL,

and barystatic and steric terms are significantly correlated (Figure 1). Barystatic and steric components render comparable contributions to GMSL changes during ENSO events (Table 1). The steric contributions reflect ocean heat storage across various depths in the Pacific Ocean (Figure 3). We offered a heuristic interpretation of the Pacific heat budget during ENSO periods in terms of diabatic exchanges at the sea surface and adiabatic redistributions within the ocean interior (Figure 4), but

more work is needed in the future to diagnose more definitively the relative contributions of surface fluxes, interbasin exchanges, vertical transports, and the deep ocean on the heat budgets. More work is also needed to understand why the anomalous GMSL response to ENSO was apparently much stronger during the 2014–2015 El Niño than during the 1997–1998 El Niño (Figure 5). Our results corroborate previous suggestions made based on models (Landerer et al., 2008) or observa-

tions during an isolated event (Dieng et al., 2014, 2015) that steric contributions to ENSO-related GMSL fluctuations are not negligible relative to barystatic contributions. These findings also have implications more generally for understanding the ocean's role in the planet's radiation imbalance and hydrological cycle.

*Acknowledgements.* The authors were supported by NASA grants NNX14AJ51G and NNH16CT00C. Helpful
conversations with Steve Nerem, Rui Ponte, Don Chambers, and John Gilson are acknowledged. The providers of the datasets are formally acknowledged in Appendix E and Table 2.

**Appendix A: Description of OLS method**

Let us regard the altimetric GMSL record (or any other data series for that matter) $Y$ for 2005–2015 (including trend and annual cycle) as a linear combination of predictors $X$,

$$Y = X\beta + \varepsilon. \tag{A1}$$

Here $X$ includes the linear trend (slope and intercept), annual cycle (sine and cosine), and MEI, $\varepsilon$ is the error term, and $\beta$ contains the regression coefficients to be solved for. The OLS estimator for $\beta$ is that vector which minimizes the variance between $Y$ and $X\beta$,

$$\hat{\beta} = MY, \tag{A2}$$

where $M \doteq \left(X^{\mathsf{T}}X\right)^{-1}X^{\mathsf{T}}$ is the Moore-Penrose pseudo-inverse and $^{\mathsf{T}}$ is matrix transpose. While OLS assumes white noise residuals, we find that $\varepsilon$ is autocorrelated (not shown). Thus, we assume



first-order autoregressive model, inflating the OLS standard errors by computing the lag-1 autocorrelation $\varphi$ and finding the effective number of data points $n^*$,

$$n^* = n\left(\frac{1-\varphi}{1+\varphi}\right), \tag{A3}$$

where here $n = 132$ months of observations over 2005–2015. This effective number of data points is then used for determining the OLS standard error for the regression coefficients,

$$\hat{\sigma}_{\hat{\beta}_j} = \sqrt{\frac{\varepsilon^{\mathsf{T}}\varepsilon}{n^*-k}\left(X^{\mathsf{T}}X\right)^{-1}_{jj}}, \tag{A4}$$

where $\hat{\beta}_j$ is the $j$th coefficient and $k = 5$ is the total number of coefficients being estimated. Similar methods are described by Chambers et al. (2012) and Calafat and Chambers (2013). Other methods

are possible for linear estimation in the presence of autocorrelated residuals (e.g., feasible generalized least squares), but we find that—in this context—these methods result in endogenous predictors (specifically, residuals of the fit are significantly correlated with the MEI predictor term), hence inconsistent estimates, and so are not employed.

**Appendix B: Evaluation of 90% confidence intervals**

All values derived from OLS regression quoted in the main text, shown in Figure 3, and given in Table 1 are 90% confidence intervals. These intervals are determined as follows. First, to account for goodness of fit, we compute the OLS standard errors, adjusting values according to the effective degrees of freedom, as above. Second, to account for uncertainty in the data, we propagate the standard errors on the data based on the OLS estimator and the usual procedures for uncertainty

propagation (e.g., Thomson and Emery, 2014),

$$\delta_{\hat{\beta}_j} = \delta_Y \sqrt{\left(MM^{\mathsf{T}}\right)_{jj}}, \tag{B1}$$

where $\delta_Y$ represents the standard error on the altimetry, GRACE, or Argo data as outlined in the text and given in Table 1. We use $\hat{\sigma}_{\hat{\beta}_j}$ and $\delta_{\hat{\beta}_j}$ to evaluate the total uncertainty $e_{\hat{\beta}_j}$,

$$e_{\hat{\beta}_j} = \sqrt{\hat{\sigma}^2_{\hat{\beta}_j} + \delta^2_{\hat{\beta}_j}}. \tag{B2}$$

Using these values for the total errors, the 90% confidence intervals are constructed as,

$$\hat{\beta}_j - t_{95} \cdot e_{\hat{\beta}_j} \le \beta_j \le \hat{\beta}_j + t_{95} \cdot e_{\hat{\beta}_j}, \tag{B3}$$

where $\beta_j$ is the true value of the $j$th coefficient and $t_{95}$ is the ninety-fifth percentile of the Student's $t$ inverse cumulative distribution given the effective degrees of freedom (Table 1).


### Appendix C:  Budgets for the annual cycle and linear trend

Here we briefly consider the GMSL budget for the annual cycle and the linear trend. These cases
have been discussed before in many previous investigations (e.g., Leuliette, 2015, and references
therein), and are discussed here mainly for the sake of completeness. Altimetry gives a GMSL trend
over 2005–2015 of $3.39 \pm 0.55$ mm yr$^{-1}$ whereas the sum of GRACE and Argo yields $3.22 \pm 0.43$
mm yr$^{-1}$ (Table 1). The residual between these two values $0.18 \pm 0.19$ mm yr$^{-1}$ is not statistically
distinguishable from zero at the 95% confidence level. We see that GRACE barystatic contributions
roughly two-thirds to the total change ($2.23 \pm 0.44$ mm yr$^{-1}$) whereas Argo steric contributes about
one-third ($0.99 \pm 0.16$ mm yr$^{-1}$). The general closure of the budget and the relative partitioning
between barystatic and steric effects is very similar to other studies for similar periods (e.g., see
Leuliette (2015) for an assessment of the observed GMSL budget for 2005–2013).

The amplitude of the GMSL annual cycle from altimetry is very similar to that from the sum of
GRACE and Argo (Table 1). Also, we notice that the barystatic and steric annual cycles are roughly
in antiphase, which leads to a GMSL annual cycle that is smaller in amplitude than the barystatic
annual cycle. This feature has been noted and discussed in numerous previous studies (e.g., Leuliette
and Miller, 2009). However, we note that, due to a slight phase difference between GMSL from
altimetry and from GRACE and Argo (Table 1), there is actually a statistically significant residual
in the annual cycle. While this is not made explicit in previous studies, it is implicit; for example,
Leuliette and Miller (2009) show a similar difference in GMSL phase between altimetry and the
sum of Argo and GRACE. It is not immediately obvious what is responsible for this discrepancy,
and it is beyond our scope to explore the issue in depth. However, we hypothesize that it is due to
sampling errors in the observing system, namely the fact that Argo does not sample at high latitudes
or, probably more importantly, on shallow continental shelf seas.

### Appendix D:  Description of Monte Carlo simulation

We evaluate what is the likelihood that the barystatic sea level term contributes more to ENSO-
related GMSL fluctuations than the steric sea level term. We make this evaluation probabilisti-
cally, performing 100,000 iterations of drawing two values, each one drawn from a separate Stu-
dent $t$-distribution. The first distribution is based on the MEI regression coefficient for the GRACE
barystatic term, with location parameter equal to the regression coefficient, scale parameter equal to
the standard error of the regression coefficient, and using the effective degrees of freedom. A draw
from this first distribution is a possible value of the barystatic contribution. Likewise, the second
distribution is based on the MEI regression coefficient for the Argo steric term, with draws from this
second distribution being possible values for the steric contribution. For each iteration, we assess the
fraction, $F = D_1/(D_1 + D_2)$, where $D_1$ and $D_2$ are the draws from the first and second distribu-
tions, respectively. Physically, $F$ represents the fractional barystatic contribution to the total GMSL



change. The histogram $P$ is derived from the realizations of $F$. Figure 2 displays the likelihood,

$$L(x) = 1 - \int_{-\infty}^{x} P(x')dx' \tag{D1}$$

where $L(x)$ is the probability (i.e., fraction of iterations) that $F > x$. For example, $L(0.6)$ is the likelihood that the barystatic term is responsible for $> 60\%$ of total GMSL change.

## Appendix E: Datasets

### E1 Satellite altimetry

The AVISO data were downloaded from the AVISO website (Table 2). The data are based on reference missions (TOPEX/Poseidon and *Jason* series) with inverted barometer correction applied, the seasonal signal retained, and glacial isostatic adjustment applied.

The CSIRO data were downloaded from the CSIRO website (Table 2). The version of the data used here had the inverse barometer and glacial isostatic adjustment corrections applied and the seasonal signals not removed ("jb_iby_srn_gtn_giy"). A 60-day smoothing was used to reduce a spurious 59-day cycle in the data related to alias of the ocean tides.

The Colorado data were downloaded from the Colorado sea level website (Table 2). The data version is version_2016rel2. A 60-day boxcar filter was also applied to the data.

The NOAA data were downloaded from the NOAA website (Table 2). The product used here is based on TOPEX/Poseidon and *Jason* series data with the seasonal signals retained. A 60-day smoothing was applied to these data and a trend of 0.3 mm yr$^{-1}$ was added to account for glacial isostatic adjustment effects not accounted for in this product.

### E2 Profiling floats

The SIO Argo data were downloaded from the SIO website (Table 2). We used the 2004–2014 climatologies with the provided monthly extensions through February of 2016.

The IPRC gridded data fields were downloaded from the IPRC website (see Table 2).

### E3 Gravimetric retrievals

The GRACE data were downloaded from Don P. Chambers' Dropbox folder (Table 2). Data gaps and missing months in these time series were filled based on cubic interpolation.

### E4 Climate indices

MEI values were downloaded from the NOAA ESRL PSD ENSO website (Table 2).





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



| | Trend (mm yr$^{-1}$) | MEI (mm MEI$^{-1}$) | Amplitude (mm) | Phase (°) | $n^*$ | $\delta_Y$ (mm) |
|---|---|---|---|---|---|---|
| $\eta$ | $3.39 \pm 0.55$ | $2.76 \pm 1.87$ | $5.74 \pm 2.40$ | $-36 \pm 24$ | 12 | 1.29 |
| $p_b + \eta_\rho$ | $3.22 \pm 0.43$ | $2.97 \pm 1.47$ | $5.81 \pm 1.90$ | $-24 \pm 19$ | 22 | 1.88 |
| $p_b$ | $2.23 \pm 0.44$ | $1.54 \pm 1.50$ | $10.04 \pm 1.92$ | $-52 \pm 11$ | 14 | 1.44 |
| $\eta_\rho$ | $0.99 \pm 0.16$ | $1.42 \pm 0.53$ | $5.61 \pm 0.68$ | $99 \pm 7$ | 49 | 1.21 |
| $\eta - p_b - \eta_\rho$ | $0.18 \pm 0.19$ | $-0.20 \pm 0.64$ | $1.26 \pm 0.82$ | $-123 \pm 37$ | 48 | 2.28 |

**Table 1.** Results of OLS applied to altimetric GMSL ($\eta$), GRACE barystatic sea level ($p_b$), Argo steric sea level ($\eta_\rho$), and linear combinations thereof. Values are given as 90% confidence intervals as described in Appendix B. Note that, while the predictors of the OLS fit include an annual sine and cosine, we present results here transformed into the amplitude and phase of a sine term using standard trigonometric transformations. Note also that $n^*$ is the effective number of data points, whereas $\delta_Y$ is the standard error evaluated for the different data as outlined in section 2.

| Dataset | Source | Location |
|---|---|---|
| Altimetry | AVISO | http://www.aviso.altimetry.fr/en/data/products/ocean-indicators-products/ |
| Altimetry | CU | http://sealevel.colorado.edu/ |
| Altimetry | NOAA | http://www.star.nesdis.noaa.gov/sod/lsa/SeaLevelRise/ |
| Altimetry | CSIRO | http://www.cmar.csiro.au/sealevel/sl_data_cmar.html |
| Argo | SIO | http://sio-argo.ucsd.edu/RG_Climatology.html |
| Argo | IPRC | http://apdrc.soest.hawaii.edu/las/v6/dataset?catitem=3 |
| GRACE | CSR | https://dl.dropboxusercontent.com/u/31563267/ocean_mass_orig.txt |
| GRACE | JPL | https://dl.dropboxusercontent.com/u/31563267/ocean_mass_orig.txt |
| GRACE | GFZ | https://dl.dropboxusercontent.com/u/31563267/ocean_mass_orig.txt |
| MEI | NOAA | http://www.esrl.noaa.gov/psd/enso/ |

**Table 2.** Locations and sources of the data used here. Websites accessible as of 2 June 2016.



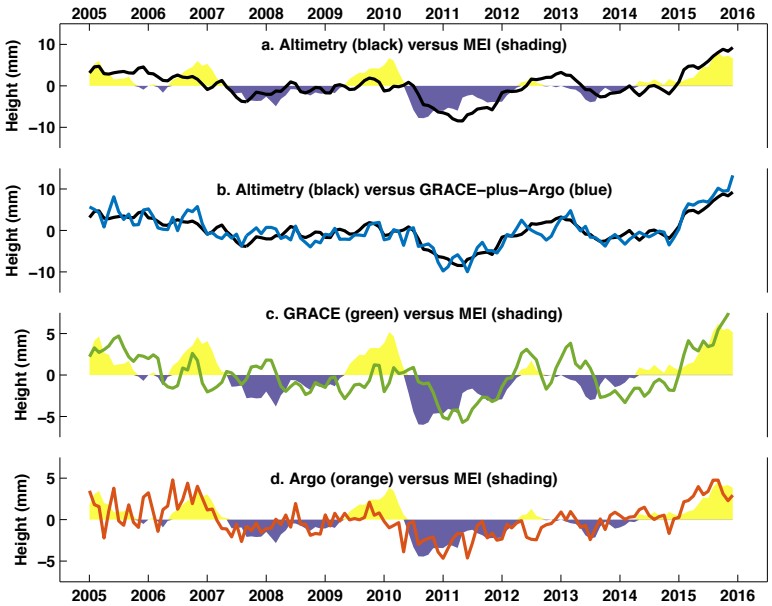

**Figure 1.** Monthly time series over 2005–2015 of **(a)** altimetric GMSL (black) and the MEI (shading), **(b)** GMSL from altimetry (black) and from GRACE and Argo (blue), **(c)** GRACE barystatic sea level (green) and the MEI (shading), and **(d)** Argo steric sea level (orange) and the MEI (shading). Linear trends and annual cycles have been removed from all time series. The MEI record has been scaled to have variance equal to that of the respective sea level time series.

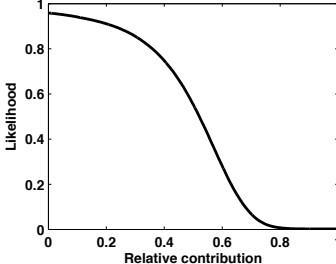

**Figure 2.** Likelihood that the barystatic contribution to ENSO-related GMSL changes exceeds a certain fraction of the sum of barystatic and steric terms based on Monte Carlo runs.




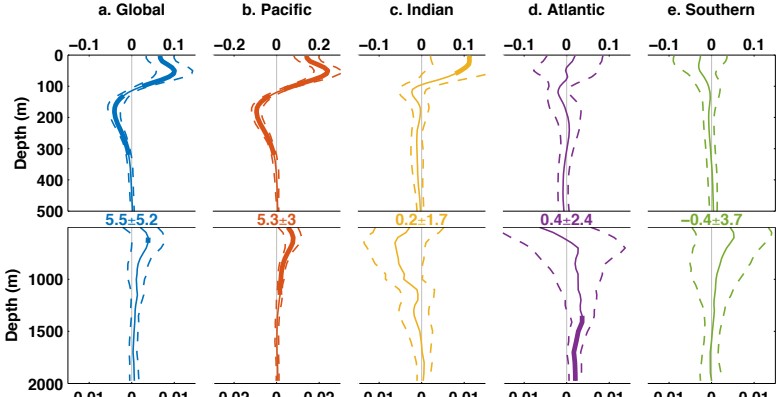

**Figure 3.** Coefficients of regressions of Argo potential temperature on the MEI (°C per MEI) over 2005–2015 over **(a)** the global ocean, **(b)** Pacific, **(c)** Indian, **(d)** Atlantic, and **(e)** Southern (south of 30°S) basins. Solid lines are the regression coefficients and dashed lines mark the 90% confidence interval. Bold lines mark significance at the 95% confidence level (i.e., one-tailed test). Note the different horizontal axis limits between the top and bottom panels. The colored values between the top and bottom panels represent the total ocean heat storage (units of ZJ per MEI; 1 ZJ ≡ $10^{21}$ J) integrated over 0–2000 m in the different basins given as 90% confidence intervals.

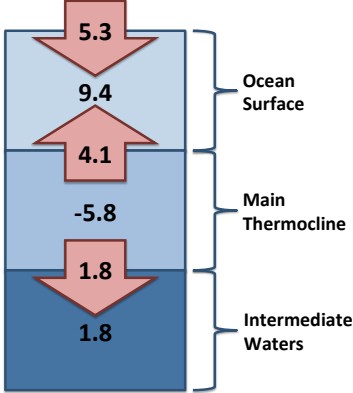

**Figure 4.** Hypothesized Pacific heat budget during El Niño events. The blue blocks are the ocean surface (0–110 m), main thermocline (120–380 m), and intermediate water (400–2000 m) layers. The red arrows are heat exchanges between the ocean layers or with overlying atmosphere. Black values are either the total ocean heat storage within the layers as given by Argo data or the required heat exchanged between them under the stated assumptions of no transports between ocean basins and no contributions from the deep (> 2000 m) ocean. Units are ZJ per unit MEI. (Note that all arrows and signs, shown here for El Niño, would be reversed for La Niña events.)



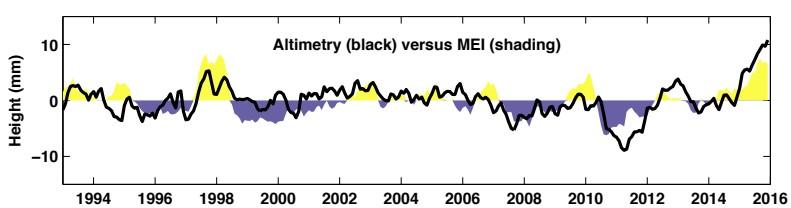

**Figure 5.** Nonseasonal anomalies of GMSL (black) and MEI (shading) over 1993–2015.