# Peer review of "El Niño, La Niña, and the global sea level budget"

_Ocean Science, 2016_

## Referee Comment (RC1) · Anonymous Referee #1 · 29 Sep 2016

This paper quantifies the relative importance of steric and barystatic contributions to global mean sea level change associated with ENSO. It is logically arranged, well-presented, concise and careful, and I hope it will be published.

I don't have any detailed comments on the text, which is very well-written. I have a few comments on aspects of the method and conclusions.

(1) Is there a possible thermosteric contribution from depths greater than 2000 m, which are not sampled by Argo? Previous studies suggest that this is non-negligible for the GMSL trend e.g. Church et al. (2011) 10.1029/2011GL048794.

(2) The method assumes the form of the predictors: MEI, constant linear trend, sinusoidal annual cycle. If the long-term variation is not a constant rate of change, the annual cycle is not sinusoidal, or the MEI is not the right measure of ENSO variation, I

suppose that the results will have a systematic error, and the conclusion might not be accurate. How well-justified are these assumptions?

(3) Did the authors consider regressing GMSL (from altimetry) against the barystatic and thermosteric contributions as predictors? In that case OLS would be inaccurate because it assumes there is no error in the independent variable, but total least squares (orthogonal regression) could be used.

(4) Having reached their conclusion that barystatic and thermosteric contributions are of comparable importance, could the authors comment on why previous authors reach different conclusions - the situation they described as "confusing" in the introduction?

(5) A minor point: it would be useful to note in Table 1 caption that n* is evaluated following Eq A3.

(6) Could Fig 5 be put as a panel in Fig 1? It would be helpful to draw attention to the difference between Figs 1a and 5. The most relevant one is that Fig 5 is the whole altimeter period. Are they different otherwise?

---

## Referee Comment (RC2) · Anonymous Referee #2 · 18 Oct 2016

The paper discusses the steric and barystatic contributions to the global mean sea level record from satellite altimetry during ENSO events. While previous studies have mainly focused on barystatic contributions, this study focuses primarily on the steric contribution to La Nina and El Nino events in sea level. The paper is well written and structured and presents some interesting ideas.

Similar to reviewer #1, I mainly have a few general questions for the authors:

1) Correlation/regression analysis: Given the complex nature of the response to ENSO particularly in barystatic sea level, I wonder if a correlation analysis directly provides conclusive results. As for example Llovel et al., 2010, Fasullo et al., 2012 elude to, the response of the barystatic sea level to ENSO events is related to the complex response of the water cycle, which includes where evaporation/precipitation is generated, what

the specific wind patterns are like, what is the setup of the hydrologic basin etc. Hence, the response in the mass part of sea level may be tied to regional variability in the extent of ENSO events as well as their strength. This makes it difficult to only use correlation and regression to quantify the response. However, for the steric part the response may be a bit more straightforward as it is mainly a warming/cooling signal of the upper ocean as this study partly also suggest. In general – as reviewer #1 also mentions – a correlation analysis can easily be misleading if one of the components is not well determined (be it by length of record or definition of indices etc.). Nevertheless, it is very interesting to see the impacts on the different layers in various ocean basins (e.g. Fig. 3, line 142ff) and think it would be great to see more details on this aspect of the study. In particular, it may be interesting to see how spatial patterns of the warming/cooling signals compare – in particular, between the different ARGO products and also compared to altimetry minus GRACE (e.g. total warming vs. layer structure).

2) Data products: Two ARGO products are being used for this study. Given the spread between data products and the focus of this paper being the steric contribution, it would be interesting to see a more detailed comparison between the two products used (or even add a third). So far, the differences in the products have mainly been evaluated to determine the error bar for the estimates but it may be worthwhile to look into the spatial distribution and spread for specific ENSO events in more detail.

3) Additional data: To add statistical significance to the steric analysis, I am wondering if the inclusion of ECCO output might be useful. The longer time series could support the correlation and regression analysis as well as basic comparisons of depths of the warming/cooing signals in the different ocean basins.
* * *

---

## Author Comment (AC1) · 28 Oct 2016

Christopher G. Piecuch and Katherine J. Quinn

cpiecuch@aer.com

Reviewer's Comment (RC): This paper quantifies the relative importance of steric and barystatic contributions to global mean sea level change associated with ENSO. It is logically arranged, well presented, concise and careful, and I hope it will be published.

RC: I don't have any detailed comments on the text, which is very well written. I have a few comments on aspects of the method and conclusions.

Authors' Response (AR): We appreciate the reviewer's positive evaluation of our paper. The manuscript will be revised accordingly, as described in the responses given below.

RC: Is there a possible thermosteric contribution from depths greater than 2000 m, which are not sampled by Argo? Previous studies suggest that this is non-negligible

for the GMSL trend e.g. Church et al. (2011) 10.1029/2011GL048794.

AR: The deep ocean's contribution to climate variability and change remains uncertain. The findings of Church et al. cited by the referee are taken from the Purkey and Johnson (2010) results, based on precise but spatiotemporally sparse hydrographic section data. Models disagree on the nature of deep ocean changes–some show warming (e.g., Song and Colberg 2011), others cooling (e.g., Wunsch and Heimbach 2014), and still others no significant thermal changes at all during recent decades (e.g., Piecuch et al. 2015). In our analysis, any deep ocean steric contributions would appear in the budget residual term, which is indistinguishable from zero (Table 1). This result is in some senses analogous to findings in Llovel et al. (2014) with regard to the deep ocean temperature trend over the 2005-2013 interval.

We will mention these topics in the paper revision, explaining that, based on our results, any contributions from un-sampled regions are indistinguishable from zero.

RC: The method assumes the form of the predictors: MEI, constant linear trend, and sinusoidal annual cycle. If the long-term variation is not a constant rate of change, the annual cycle is not sinusoidal, or the MEI is not the right measure of ENSO variation, I suppose that the results will have a systematic error, and the conclusion might not be accurate. How well justified are these assumptions?

AR: A degree of subjectivity in model selection is inevitable and unavoidable. We believe that the form of the predictors assumed here is reasonable judging from previous works (cited in the introduction). Regression onto these parameters explains 96% (99%) of the monthly variance in the altimetric sea level record over 2005-2015 (1993-2015), and the regression coefficients are all significant, suggesting that our assumptions are justified. Using indices other than the MEI, or allowing lags between MEI and GMSL, yields similar results. Variations in the GMSL annual cycle or its long-term rate of change are of course possible but are not obvious from the altimetry data, and addressing these issues would require a more detailed and dedicated study beyond

the scope of our analysis.

In the revision, we will argue more clearly that our assumptions are justified.

RC: Did the authors consider regressing GMSL (from altimetry) against the barystatic and thermosteric contributions as predictors? In that case OLS would be inaccurate because it assumes there is no error in the independent variable, but total least squares (orthogonal regression) could be used.

AR: We appreciate being made aware of the method of total least squares for the case that the predictors have errors. For various reasons, we hesitate to regress GMSL onto barystatic and thermosteric terms, as suggested by the reviewer. From a mathematical perspective, such a regression would be problematic, because, as we show in the paper (Fig. 1), barystatic and thermosteric terms are correlated. Thus, the regressors would not be linearly independent, as required by least squares. Further, and notwithstanding correlation between the regressors, such regression would be physically unenlightening; from the hydrostatic relation (cf. Eqn. 2.11 in Gill and Niiler 1973), it must be that the coefficients of such a regression equal one, and hence there is insufficient motivation to perform the additional analyses suggested by the reviewer.

For these reasons, we have not made any changes to the paper on these points.

RC: Having reached their conclusion that barystatic and thermosteric contributions are of comparable importance, could the authors comment on why previous authors reach different conclusions—the situation they described as "confusing" in the introduction

AR: There are a few potential reasons for this confusion, some of which are given below:

The nature of GMSL changes linked to ENSO has been inferred from observations of isolated events, such as the 2010/2011 La Niña. These particular events might not be representative of the general GMSL response to ENSO. As revealed by Fasullo et al. (2013), isolated GMSL events can be related not only to ENSO but also, for example,

IOD and SAM. These considerations complicate interpretation of GMSL, barystatic, and thermosteric data for isolated events in terms of GMSL response during ENSO events more generally.

Some studies base conclusions regarding barystatic contributions to changes in GMSL on strong correlation between the two signals. But, correlations only take into account the relative phase of the signals, and not their relative magnitudes. This can paint a deceptive picture in the present context. To give a toy example, consider two time series, x(t) = sin(t), and y(t) = 2sin(t). Obviously, x and y are perfectly correlated, but the changes in x do not entirely account for changes in y, as the former only has half the magnitude of the latter. Thus, in cases such as the present, where all terms (i.e., GSML, steric, barystatic) exhibit similar phase behavior, but varying magnitude, a more thorough analysis must consider both phasing and magnitude of the signals.

Observational and modeling products used to evaluate the steric and barystatic effects on sea level changes are of course characterized by errors. As we reveal in our accompanying response to Referee #2, Argo grids processed by different centers can show important differences regionally and globally. These errors in models and data could lead to links between GMSL and its components that are too strong or weak.

Language will be added to the revised introduction and conclusion that speaks to these points without unduly criticizing previous works.

RC: A minor point: it would be useful to note in Table 1 caption that n* is evaluated following Eq. A3.

AR: We will certainly make this note in the revised manuscript.

RC: Could Fig 5 be put as a panel in Fig 1? It would be helpful to draw attention to the difference between Figs 1a and 5. The most relevant one is that Fig 5 is the whole altimeter period. Are they different otherwise?

AR: While it's admittedly a matter of subjective, aesthetic preference, we are hesitant

to add Figure 5a as a panel in Figure 1, since the former covers a time period different from that in the other Figure 1 panels. Nevertheless, we will add text into the manuscript that points out the differences between the two figures. These differences include the period of display, as pointed out by the reviewer, and also the facts that the (removed) annual cycle and linear trend are estimated for the 2005-2015 period in Figure 1 while they are estimated for the 1993-2015 period in Figure 5a.

References

Fasullo et al. (2013), Geophys. Res. Lett., 40, 4368-4373, doi:10.1002/grl.50834.

Gill and Niiler (1973), Deep Sea Res., 20, 141-177.

Piecuch et al. (2015), Ocean Model., 96, 214-220.

Purkey and Johnson (2010), J. Climate, 23, 6336-6351.

Song and Colberg (2011), J. Geophys. Res., 116, C02020, doi:10.1029/2010JC006601.

Wunsch and Heimbach (2014), J. Phys. Oceanogr., 44, 2013-2030.

---

## Author Comment (AC2) · 28 Oct 2016

Christopher G. Piecuch and Katherine J. Quinn

cpiecuch@aer.com

Reviewer's Comment (RC): The paper discusses the steric and barystatic contributions to the global mean sea level record from satellite altimetry during ENSO events. While previous studies have mainly focused on barystatic contributions, this study focuses primarily on the steric contribution to La Nina and El Nino events in sea level. The paper is well written and structured and presents some interesting ideas. Similar to reviewer #1, I mainly have a few general questions for the authors.

Authors' Response (AR): We appreciate the reviewer's positive evaluation of our paper. The manuscript will be revised accordingly, as described in the responses given below.

RC: Correlation/regression analysis: Given the complex nature of the response to ENSO particularly in barystatic sea level, I wonder if a correlation analysis directly

provides conclusive results. As for example Llovel et al., 2010, Fasullo et al., 2012 allude to, the response of the barystatic sea level to ENSO events is related to the complex response of the water cycle, which includes where evaporation/precipitation is generated, what the specific wind patterns are like, what is the setup of the hydrologic basin etc. Hence, the response in the mass part of sea level may be tied to regional variability in the extent of ENSO events as well as their strength. This makes it difficult to only use correlation and regression to quantify the response. However, for the steric part the response may be a bit more straightforward as it is mainly a warming/cooling signal of the upper ocean as this study partly also suggest. In general – as reviewer #1 also mentions – a correlation analysis can easily be misleading if one of the components is not well determined (be it by length of record or definition of indices etc.). Nevertheless, it is very interesting to see the impacts on the different layers in various ocean basins (e.g. Fig. 3, line 142ff) and think it would be great to see more details on this aspect of the study. In particular, it may be interesting to see how spatial patterns of the warming/cooling signals compare – in particular, between the different ARGO products and also compared to altimetry minus GRACE (e.g. total warming vs. layer structure).

AR: There are many warranted concerns being voiced here. Some are already addressed in the manuscript. We acknowledge that, due to the short duration of the data records, some of the relationships seen here may be specific to the time period studied, and not representative of 'the' GMSL response during ENSO (cf. lines 189-191).

We admit that more details on the spatial patterns of steric changes would be of interest to the reader. While the difference between altimetry and GRACE is a vertically integrated measure, and does not give insight onto vertical structure, some analyses along the lines suggested by the reviewer are possible. For example, comparing steric changes globally and regionally from the two Argo centers (Scripps and IPRC) would be straightforward to perform and interpret (i.e., in terms of differing data processing strategies between the two centers), and will be included in the revised manuscript

(see immediately below).

RC: Data products: Two ARGO products are being used for this study. Given the spread between data products and the focus of this paper being the steric contribution, it would be interesting to see a more detailed comparison between the two products used (or even add a third). So far, the differences in the products have mainly been evaluated to determine the error bar for the estimates but it may be worthwhile to look into the spatial distribution and spread for specific ENSO events in more detail.

AR: Given our focus on steric contributions, assessment of uncertainties in various steric products is important. A comprehensive assessment is beyond our scope, better left to a more technical dedicated manuscript, but we will give a few preliminary 'case examples' comparing Scripps and IPRC during particular events. As shown in the figures below, the two products differ noticeably in terms of anomalous regional temperature and global steric changes during the recent El Niño (e.g., last six months of 2015, July-December).

We will include similar figures in the revised paper, point out the discrepancies, and encourage a more thorough future assessment.

RC: Additional data: To add statistical significance to the steric analysis, I am wondering if the inclusion of ECCO output might be useful. The longer time series could support the correlation and regression analysis as well as basic comparisons of depths of the warming/cooling signals in the different ocean basins.

AR: We agree that considering an ECCO solution would allow analysis of a longer period (1992-present). It would also facilitate a more detailed mechanistic understanding of the processes contributing to the global and regional steric changes. Yet, such consideration would make for a much longer paper with a considerably different scope. We think that there is value in providing a concise "first analysis" of the GMSL budget related to ENSO events based purely on observations.

In the discussion of the revision, we will point more explicitly to an analysis along the lines suggested by the reviewer as a logical "next step" in the investigation of GMSL changes linked to ENSO variability.

[Figure]

[Figure]

**Fig. 1.** Time series of nonseasonal anomalous thermosteric sea level from: average of SIO and IPRC products (black); SIO product (dark gray); and IPRC data (light gray).

[Figure]

**Fig. 2.** (A) Nonseasonal anomalous steric sea level averaged over July-December 2015 based on SIO gridded data. (B) As in (A) but computed using gridded data from IPRC. (C) The difference (A)-(B).

---

## Author Response (AR1)

**(1)**: This paper quantifies the relative importance of steric and barystatic contributions
to global mean sea level change associated with ENSO. It is logically arranged, well
presented, concise and careful, and I hope it will be published.

I don't have any detailed comments on the text, which is very well written. I have a few
comments on aspects of the method and conclusions.

**(2)**: We appreciate the reviewer's positive evaluation of our paper. The manuscript has
been revised accordingly, as described in the responses given below.

**(1)**: Is there a possible thermosteric contribution from depths greater than 2000 m,
which are not sampled by Argo? Previous studies suggest that this is non-negligible for
the GMSL trend e.g. Church et al. (2011) 10.1029/2011GL048794.

**(2)**: The deep ocean's contribution to climate variability and change remains uncertain.
The findings of Church et al. cited by the referee are taken from the Purkey and Johnson
(2010) results, based on precise but spatiotemporally sparse hydrographic section data.
Models disagree on the nature of deep ocean changes—some show warming (e.g., Song
and Colberg 2011), others cooling (e.g., Wunsch and Heimbach 2014), and still others no
significant thermal changes at all during recent decades (e.g., Piecuch et al. 2015). In our
analysis, any deep ocean steric contributions would appear in the budget residual term,
which is indistinguishable from zero (Table 1). This result is in some senses analogous to
findings in Llovel et al. (2014) with regard to the deep ocean temperature trend over the
2005-2013 interval.

**(3)** We mention these topics in the paper revision, explaining that, based on our results,
any contributions from un-sampled regions are indistinguishable from zero (see lines
124-127 in the tracked-changes manuscript below).

**(1)**: The method assumes the form of the predictors: MEI, constant linear trend, and
sinusoidal annual cycle. If the long-term variation is not a constant rate of change, the annual cycle is not sinusoidal, or the MEI is not the right measure of ENSO variation, I
suppose that the results will have a systematic error, and the conclusion might not be
accurate. How well justified are these assumptions?

**(2)**: A degree of subjectivity in model selection is inevitable and unavoidable. We believe
that the form of the predictors assumed here is reasonable judging from previous works
(cited in the introduction). Regression onto these parameters explains 96% (99%) of the
monthly variance in the altimetric sea level record over 2005-2015 (1993-2015), and the
regression coefficients are all significant, suggesting that our assumptions are justified.
Using indices other than the MEI, or allowing lags between MEI and GMSL, yields similar
results. Variations in the GMSL annual cycle or its long-term rate of change are of course
possible but are not obvious from the altimetry data, and addressing these issues would
require a more detailed and dedicated study beyond the scope of our analysis.

(**3**) In the revision, we argue more clearly that our assumptions are justified (lines 109-
113 in the tracked-changes version below).

**(1)**: Did the authors consider regressing GMSL (from altimetry) against the barystatic
and thermosteric contributions as predictors? In that case OLS would be inaccurate
because it assumes there is no error in the independent variable, but total least squares
(orthogonal regression) could be used.

**(2)**: We appreciate being made aware of the method of total least squares for the case
that the predictors have errors. For various reasons, we hesitate to regress GMSL onto
barystatic and thermosteric terms, as suggested by the reviewer. From a mathematical
perspective, such a regression would be problematic, because, as we show in the paper
(Fig. 1), barystatic and thermosteric terms are correlated. Thus, the regressors would
not be linearly independent, as required by least squares. Further, and notwithstanding
correlation between the regressors, such regression would be physically unenlightening;
from the hydrostatic relation (cf. Eqn. 2.11 in Gill and Niiler 1973), it must be that the
coefficients of such a regression equal one, and hence there is insufficient motivation to
perform the additional analyses suggested by the reviewer.

(**3**) For these reasons, we have not made any changes to the paper on these points.

**(1)**: Having reached their conclusion that barystatic and thermosteric contributions are
of comparable importance, could the authors comment on why previous authors reach
different conclusions—the situation they described as "confusing" in the introduction

**(2)**: There are a few potential reasons for this confusion, some of which are given below:

The nature of GMSL changes linked to ENSO has been inferred from observations
of isolated events, such as the 2010/2011 La Niña. These particular events might not be
representative of the general GMSL response to ENSO. As revealed by Fasullo et al.
(2013), isolated GMSL events can be related not only to ENSO but also, for example, IOD

and SAM. These considerations complicate interpretation of GMSL, barystatic, and thermosteric data for isolated events in terms of GMSL response during ENSO events more generally.

Some studies base conclusions regarding barystatic contributions to changes in GMSL on strong correlation between the two signals. But, correlations only take into account the relative phase of the signals, and not their relative magnitudes. This can paint a deceptive picture in the present context. To give a toy example, consider two time series, x(t) = sin(t), and y(t) = 2sin(t). Obviously, x and y are perfectly correlated, but the changes in x do not entirely account for changes in y, as the former only has half the magnitude of the latter. Thus, in cases such as the present, where all terms (i.e., GSML, steric, barystatic) exhibit similar phase behavior, but varying magnitude, a more thorough analysis must consider both phasing and magnitude of the signals.

Observational and modeling products used to evaluate the steric and barystatic effects on sea level changes are of course characterized by errors. For instance, as we show below in our response to Referee #2, Argo grids processed by different centers can show important differences regionally and globally. These errors in models and data could lead to links between GMSL and its components that are too strong or weak.

(**3**) Language has been added to the revised introduction and conclusion that speaks to these points without unduly criticizing previous works (lines 36-39 and 228-231 in the tracked-changes version of the manuscript).

(**1**): A minor point: it would be useful to note in Table 1 caption that n* is evaluated following Eq. A3.

(**2**): We will certainly make this note in the revised manuscript.

(**3**) See Caption to Table 1 in the tracked changes version of the paper below.

(**1**): Could Fig 5 be put as a panel in Fig 1? It would be helpful to draw attention to the difference between Figs 1a and 5. The most relevant one is that Fig 5 is the whole altimeter period. Are they different otherwise?

(**2**): While it's admittedly a matter of subjective, aesthetic preference, we are hesitant to add Figure 5a as a panel in Figure 1, since the former covers a time period different from that in the other Figure 1 panels. Nevertheless, we will add text into the manuscript that points out the differences between the two figures. These differences include the period of display, as pointed out by the reviewer, and also the facts that the (removed) annual cycle and linear trend are estimated for the 2005-2015 period in Figure 1 while they are estimated for the 1993-2015 period in Figure 5a.

(**3**) See lines 218-220 in the revised tracked changes version below.

*Response to Interactive comment on "El Niño, La Niña, and the global sea level budget"*
*by Anonymous Referee #2*

**(1)**: The paper discusses the steric and barystatic contributions to the global mean sea
level record from satellite altimetry during ENSO events. While previous studies have
mainly focused on barystatic contributions, this study focuses primarily on the steric
contribution to La Nina and El Nino events in sea level. The paper is well written and
structured and presents some interesting ideas. Similar to reviewer #1, I mainly have a
few general questions for the authors.

**(2)**: We appreciate the reviewer's positive evaluation of our paper. The manuscript has
been revised accordingly, as described in the responses given below.

**(1)**: Correlation/regression analysis: Given the complex nature of the response to ENSO
particularly in barystatic sea level, I wonder if a correlation analysis directly provides
conclusive results. As for example Llovel et al., 2010, Fasullo et al., 2012 allude to, the
response of the barystatic sea level to ENSO events is related to the complex response
of the water cycle, which includes where evaporation/precipitation is generated, what
the specific wind patterns are like, what is the setup of the hydrologic basin etc. Hence,
the response in the mass part of sea level may be tied to regional variability in the
extent of ENSO events as well as their strength. This makes it difficult to only use
correlation and regression to quantify the response. However, for the steric part the
response may be a bit more straightforward as it is mainly a warming/cooling signal of
the upper ocean as this study partly also suggest. In general – as reviewer #1 also
mentions – a correlation analysis can easily be misleading if one of the components is
not well determined (be it by length of record or definition of indices etc.).
Nevertheless, it is very interesting to see the impacts on the different layers in various
ocean basins (e.g. Fig. 3, line 142ff) and think it would be great to see more details on
this aspect of the study. In particular, it may be interesting to see how spatial patterns
of the warming/cooling signals compare – in particular, between the different ARGO
products and also compared to altimetry minus GRACE (e.g. total warming vs. layer
structure).

**(1)**: There are many warranted concerns being voiced here. Some are already addressed
in the manuscript. We acknowledge that, due to the short duration of the data records,
some of the relationships seen here may be specific to the time period studied, and not
representative of 'the' GMSL response during ENSO (cf. lines 220-224 below).

We admit that more details on the spatial patterns of steric changes would be of
interest to the reader. While the difference between altimetry and GRACE is a vertically
integrated measure, and does not give insight onto vertical structure, some analyses
along the lines suggested by the reviewer are possible. For example, comparing steric
changes globally and regionally from the two Argo centers (Scripps and IPRC) would be
straightforward to perform and interpret (i.e., in terms of differing data processing
strategies between the two centers), and will be included in the revised manuscript (see immediately below).

**(1)**: Data products: Two ARGO products are being used for this study. Given the spread
between data products and the focus of this paper being the steric contribution, it
would be interesting to see a more detailed comparison between the two products used
(or even add a third). So far, the differences in the products have mainly been evaluated
to determine the error bar for the estimates but it may be worthwhile to look into the
spatial distribution and spread for specific ENSO events in more detail.

**(2)**: Given our focus on steric contributions, assessment of uncertainties in various steric
products is important. A comprehensive assessment is beyond our scope, better left to a
more technical dedicated manuscript, but we will give a few preliminary 'case examples'
comparing Scripps and IPRC during particular events. As shown in the figures below, the
two products differ noticeably in terms of anomalous regional temperature and global
steric changes during the recent El Niño (e.g., last six months of 2015, July-December).

(**3**) We include similar figures in the revised paper (see edited Figure 1a and new Figure
5), point out the discrepancies, and encourage a more thorough future assessment (see
lines 119-215 in the revised tracked changes version below).

**(1)**: Additional data: To add statistical significance to the steric analysis, I am wondering
if the inclusion of ECCO output might be useful. The longer time series could support the
correlation and regression analysis as well as basic comparisons of depths of the
warming/cooling signals in the different ocean basins.

**(2)**: We agree that considering an ECCO solution would allow analysis of a longer period
(1992-present). It would also facilitate a more detailed mechanistic understanding of the
processes contributing to the global and regional steric changes. Yet, such consideration
would make for a much longer paper with a considerably different scope. We think that
there is value in providing a concise "first analysis" of the GMSL budget related to ENSO
events based purely on observations.

(**3**) In the discussion of the revision, we point more explicitly to an analysis along the
lines suggested by the reviewer as a logical "next step" in the investigation of GMSL
changes linked to ENSO variability (see lines 172-176 in tracked changes version below).

**Figures**

[Figure]

**Figure R1**: Time series of nonseasonal anomalous thermosteric sea level from: average
of SIO and IPRC products (black); SIO product (dark gray); and IPRC data (light gray).

[Figure]

**Figure R2**: (A) Spatial map of nonseasonal anomalous steric sea level averaged over the
last six monthly of 2015 (July-December) based on SIO gridded data. (B) As in (A) but
computed using gridded data from IPRC. (C) Difference (A) minus (B).

[revised manuscript text omitted]